# Antibiotic use and prescription and its effects on *Enterobacteriaceae* in the gut in children with mild respiratory infections in Ho Chi Minh City, Vietnam. A prospective observational outpatient study

**Ngo Ngoc Quang Minh**[1,2], **Pham Van Toi**[1], **Le Minh Qui**[2], **Le Binh Bao Tinh**[2], **Nguyen Thi Ngoc**[2], **Le Thi Ngoc Kim**[2], **Nguyen Hanh Uyen**[1], **Vu Thi Ty Hang**[1], **Nguyen Thi Thuy Chinh B'Krong**[1], **Nguyen Thi Tham**[1], **Thai Dang Khoa**[1], **Huynh Duy Khuong**[1], **Pham Quynh Vi**[1], **Nguyen Ngoc Hong Phuc**[1], **Le Thi Minh Vien**[1], **Thomas Pouplin**[1], **Doan Van Khanh**[1¤a], **Pham Nguyen Phuong**[1], **Phung Khanh Lam**[1], **Heiman F. L. Wertheim**[1,3¤b], **James I. Campbell**[1], **Stephen Baker**[1], **Christopher M. Parry**[4,5], **Juliet E. Bryant**[1¤c], **Constance Schultsz**[1,6], **Nguyen Thanh Hung**[2], **Menno D. de Jong**[1,7☺], **H. Rogier van Doorn**[1,3☺]*

1 Oxford University Clinical Research Unit, Hospital for Tropical Diseases, Ho Chi Minh City, Vietnam, 2 Children's Hospital 1, Ho Chi Minh City, Vietnam, 3 Nuffield Department of Clinical Medicine, Centre for Tropical Medicine and Global Health, University of Oxford, Henry Wellcome Building for Molecular Physiology, Old Road Campus, Headington, Oxford, United Kingdom, 4 Clinical Sciences, Liverpool School of Tropical Medicine, Pembroke Pl, Liverpool, United Kingdom, 5 School of Tropical Medicine and Global Health, Nagasaki University, Nagasaki, Japan, 6 Department of Global Health-Amsterdam, Institute of Global Health and Development, Amsterdam University Medical Centres, University of Amsterdam, Amsterdam, The Netherlands, 7 Department of Medical Microbiology, Amsterdam University Medical Centres, Amsterdam, The Netherlands

☺ These authors contributed equally to this work.
¤a Current address: Faculty of Medicine, Tan Tao University, Tan Duc, Duc Hoa, Long An, Vietnam
¤b Current address: Department of Medical Microbiology, Radboud University Medical Centre, Nijmegen, The Netherlands
¤c Current address: Mérieux Foundation, Lyon, France
* rvandoorn@oucru.org

**Data Availability Statement:** All relevant data are within the manuscript and its Supporting Information files.

## Abstract

### Background and objectives

Treatment guidelines do not recommend antibiotic use for acute respiratory infections (ARI), except for streptococcal pharyngitis/tonsillitis and pneumonia. However, antibiotics are prescribed frequently for children with ARI, often in absence of evidence for bacterial infection. The objectives of this study were 1) to assess the appropriateness of antibiotic prescriptions for mild ARI in paediatric outpatients in relation to available guidelines and detected pathogens, 2) to assess antibiotic use on presentation using questionnaires and detection in urine 3) to assess the carriage rates and proportions of resistant intestinal *Enterobacteriaceae* before, during and after consultation.

**Funding:** This work was supported by the Wellcome Trust of Great Britain [077078/Z/05/Z and 089276/Z/09/Z]. The funders had no role in the study design; in the collection, analysis, and interpretation of data; in the writing of the report; and in the decision to submit the article for publication.

**Competing interests:** The authors have declared that no competing interests exist.

## Materials and methods

Patients were prospectively enrolled in Children's Hospital 1, Ho Chi Minh City, Vietnam and diagnoses, prescribed therapy and outcome were recorded on first visit and on follow-up after 7 days. Respiratory bacterial and viral pathogens were detected using molecular assays. Antibiotic use before presentation was assessed using questionnaires and urine HPLC. The impact of antibiotic usage on intestinal *Enterobacteriaceae* was assessed with semi-quantitative culture on agar with and without antibiotics on presentation and after 7 and 28 days.

## Results

A total of 563 patients were enrolled between February 2009 and February 2010. Antibiotics were prescribed for all except 2 of 563 patients. The majority were 2nd and 3rd generation oral cephalosporins and amoxicillin with or without clavulanic acid. Respiratory viruses were detected in respiratory specimens of 72.5% of patients. Antibiotic use was considered inappropriate in 90.1% and 67.5%, based on guidelines and detected pathogens, respectively. On presentation parents reported antibiotic use for 22% of patients, 41% of parents did not know and 37% denied antibiotic use. Among these three groups, six commonly used antibiotics were detected with HPLC in patients' urine in 49%, 40% and 14%, respectively. Temporary selection of 3rd generation cephalosporin resistant intestinal *Enterobacteriaceae* during antibiotic use was observed, with co-selection of resistance to aminoglycosides and fluoroquinolones.

## Conclusions

We report overuse and overprescription of antibiotics for uncomplicated ARI with selection of resistant intestinal *Enterobacteriaceae*, posing a risk for community transmission and persistence in a setting of a highly granular healthcare system and unrestricted access to antibiotics through private pharmacies.

## Registration

This study was registered at the International Standard Randomised Controlled Trials Number registry under number ISRCTN32862422: http://www.isrctn.com/ISRCTN32862422

## Introduction

Antimicrobial resistance is a major threat to global health, sustainable development and the global economy [1–3]. Globally, the number of deaths related to antimicrobial resistance is predicted to increase from 700,000 currently to 10 million by 2050 without intervention [1]. In the US and the EU, 23,000 and 25,000 deaths secondary to drug-resistant infections are estimated to occur annually, respectively [4, 5]. These numbers are 3–5 times higher (relative to population size) in Thailand, a middle-income country where antibiotic use is relatively well controlled [6]. Figures are likely to be higher in settings with unrestricted access, a highly granular healthcare system and lack of antibiotic stewardship programmes like Vietnam, where antibiotic usage and resistance rates are among the highest in Asia [7, 8].

Antibiotics are commonly used or prescribed for mild acute respiratory infections (ARI) in both children and adults in primary care. In the UK, for instance, a quarter of the population visit their general practitioner for ARI annually, accounting for 60% of total GP antibiotic prescribing [9, 10]. In Vietnam, the 14th most populous country globally and with a population of 90 million and a high infectious diseases burden, antibiotics are readily available over-the-counter without prescription, including in community health posts and from private pharmacies [11]. Previous work has shown that 78% of antibiotics for human use were dispensed in the 60,000 private pharmacies, of which around 90% without prescription, and that the most common indication was for ARI (80%) [12]. In a cross-sectional study in northern Vietnam 75% of children under 5 had received antibiotics for an ARI in the preceding month, the large majority dispensed without prescription in private pharmacies [13]. However, aetiological studies show that ARIs are mostly caused by viruses and clinical trials have shown no efficacy of antibiotics in treating conditions like acute bronchitis, bronchiolitis and (naso)pharyngitis [14–17].

To inform policy and potential interventions, we aimed to quantify real life antibiotic prescription rates compared with what was recommended from clinical guidelines and in relation to detected pathogens in patients presenting with ARI at an outpatient clinic at a large tertiary care paediatric hospital (Children's Hospital 1) in Ho Chi Minh City, Vietnam. We also assessed antibiotic use on presentation using questionnaires and HPLC detection of 6 frequently prescribed antibiotics in urine (S1 Table), and the immediate impact of antibiotic usage on selection of resistant intestinal *Enterobacteriaceae* using semi-quantitative culture.

## Materials and methods

### Patients and samples

Patients were enrolled at the respiratory infection examination rooms at the outpatient clinic of Children's hospital 1 (CH1) in Ho Chi Minh City, Vietnam. CH1 is a large tertiary referral centre with an outpatient clinic with 54 examination rooms where 1,600,000 patients are seen annually. At the 2 respiratory infection examination rooms, 45,000 patients are seen annually. Patients are referred to the respiratory infection examination rooms when they register at the outpatient department by nursing staff when presenting with respiratory symptoms as the chief complaint. Patients were eligible for enrolment if they were under 16 years of age, had a clinical diagnosis of ARI (defined as having an acute onset, within the last 5 days, of at least one of the following symptoms as the chief complaint: cough, sore throat, runny nose or nasal congestion), had no underlying illness (except asthma), were not admitted to hospital, and agreed to return for a follow up visit after 1 week. After interim analysis of the first 100 patients a strong increment in the proportions of resistant counted colonies was shown. A study amendment was made and approved and the remaining patients to be enrolled were asked to return after 4 weeks for an additional rectal swab to assess the duration of this effect. Patients with a clinical diagnosis of sinusitis or acute otitis media were not enrolled. Written informed consent was obtained from parents or legal guardians. Eligible patients were recruited by a study doctor during a one-hour period on Mondays, Tuesdays, Wednesdays and Thursdays, with weekly enrolment restricted to a maximum of 10 to 12 patients. Apart from follow-up visits for study purposes, it was common for patients to return a number of times in the first week after first presentation as part of standard outpatient care.

Demographic, clinical and antibiotic prescription and usage data were collected in case report forms (CRF). Attending physicians assessed and recorded clinical diagnosis and outcome. At enrolment combined nose and throat swabs (NTS) in Viral Transport Medium, a

rectal swab in normal saline and a urine sample were collected. At follow-up visits after 1 week and, in a subset of patients, after 4 weeks an additional rectal swab was collected.

Based on systematic reviews, local hospital guidelines and international guidelines antibiotics were considered inappropriate based on clinical diagnosis if prescribed for a clinical diagnosis of uncomplicated ARI other than pneumonia or streptococcal pharyngitis/tonsillitis, and based on clinical diagnosis and molecular diagnostic results if in addition to the clinical diagnosis a virus was detected (and no bacteria) [14–16, 18–20].

### Ethics approval

Protocols and amendments were reviewed and approved by the Institutional Review Boards of Children's Hospital 1 and the Health Service of Ho Chi Minh City, and by the University of Oxford Tropical Research Ethics Committee (OxTREC).

### Molecular diagnostics

The following bacterial and viral pathogens were detected using inhouse molecular diagnostics: *Streptococcus pneumoniae*, *Haemophilus influenzae* type b, *Mycoplasma pneumoniae*, *Legionella pneumophila*, *Chlamydophila pneumoniae* and *psittaci*, *Bordetella pertussis* and *parapertussis*, Influenza virus A-B, Parainfluenza virus 1–4, Respiratory Syncytial Virus A-B, human metapneumovirus A-B, rhinoviruses, enteroviruses, parechoviruses, coronaviruses, bocaviruses and adenoviruses. All assays were published previously, except the *L. pneumophila* assay [21–28]. For *L. pneumophila* an inhouse validated assay, developed and used in the Amsterdam University Medical Centres, was used. Nucleic acids were extracted using an automated commercial Guanidinium Thiocyanate (GuSCN) based method on the EasyMag (Easy MAG 2.0, bioMérieux, Marcy l'Étoile, France) or MagnaPure 96 (Roche, Mannheim, Germany) platform. Viral RNA was reversely transcribed using Superscript III reverse transcriptase (Invitrogen, Carlsbad, California, USA) and random hexamers (Roche). Real-time PCR was performed on a LC480 II Thermocycler (Roche). Primers were manufactured by Sigma Proligo (Singapore). Probes containing Minor Groove Binders (MGB) were manufactured by Applied Biosystems Inc. (Foster City, CA, USA), probes containing LCRED 610, 670 or CYAN500 fluorophores or with incorporated Locked Nucleic Acid (LNA) residues were manufactured by Tib Molbiol (Berlin, Germany), and probes containing only HEX or FAM by Sigma Proligo. Equine Arteritis Virus and Phocid Herpes Virus were used as non-competitive internal controls in PCRs with RNA and DNA targets, respectively [29, 30]. Testing was not part of standard of care, was conducted batch-wise in a research laboratory and results were not reported to the treating physicians.

### Measurement of antibiotic levels in urine

Antibiotic levels in urine were determined using validated in-house protocols, as described earlier [31]. The most frequently used antibiotics in the hospital and sold in the pharmacy of Children's Hospital 1 are listed in S1 Table. Briefly, urine concentrations of six frequently used antibiotics (cephadroxil, cephalexin, cefaclor, cefixime, amoxicillin and cefuroxime) were assessed using in-house validated High-Performance Liquid Chromatography (HPLC) protocols. Briefly, HPLC was done on a Lachrom Elite–Hitachi HPLC controlled by EZChrom v3.18 software (Merck–Hitachi Japan). Solid phase extraction was done using Isolute 96 fixed well plates (Biotaga AB, Uppsala, Sweden). Separation was done through a 5μm LichroCart 240x4.6mm Purosphere RP-18/5μm 4x4 RP-18e for cephadroxil, cefaclor, cephalexin and cefixime (limit of detection: 0.08mg/L; lower limit of quantification [LLOQ]: 0.3mg/L) and

through a 5μm 125x4mm RP-8/5μm 4x4 RP-8e for amoxicillin and cefuroxime (LOD: 0.1mg/L and 0.05mg/L, respectively; LLOQ: 0.2mg/L) [18].

## Semi-quantitative culture of *Enterobacteriaceae*

Rectal swabs were kept at 4˚C and processed within 24 hours of collection. Swabs were suspended in saline and serially diluted. Dilutions were cultured on MacConkey agar and the dilution yielding between 20–200 colonies was made again the next day from the original specimen for culture on MacConkey plates containing tetracycline (4 mg/L), amoxicillin (8 mg/L), amoxicillin-clavulanic acid (8–4 mg/L), ceftazidime (2 mg/L), ciprofloxacin (1 mg/L), trimethoprim/sulfamethoxazole (2/38 mg/L), gentamicin (4 mg/L), meropenem (4 mg/L) and a control plate without antibiotics. The concentrations used were selected based on the CLSI criteria for intermediate susceptibility of these antibiotics for *Enterobacteriaceae*. Pan-susceptible and resistant ATCC isolates were used as internal quality controls for each batch of plates. Lactose fermenting *Enterobacteriaceae* were identified as large circular smooth pink colonies by experienced clinical microbiology laboratory technicians and were counted. For each antibiotic the proportion of resistant *Enterobacteriaceae* was determined as the ratio of bacterial counts on plates with and without different antibiotics. Only pink (lactose fermenting) colonies were counted and throughout this manuscript when we refer to *Enterobacteriaceae* counts, lactose fermenting *Enterobacteriaceae* are meant. Bacterial counts (+1) were logarithmically transformed for display in graphs and regression was used to calculate the position of lines representing the mean for each collection timepoint. This semi-quantitative method was designed to be a simple and elegant way to show relative selection / enrichment of resistant easily culturable relevant human pathogens during appropriate or inappropriate use of antibiotics for an infection in a different body compartment.

## Statistical analysis

All variables of interest were summarised by group (age, clinical diagnosis, single infection or co-infection). For descriptive statistics, prevalence and percentage were used for categorical variables while mean and standard deviation or median and interquartile range (IQR) were used for normally or non-normally distributed continuous data, respectively. Comparisons of epidemiological and clinical characteristics among age groups or infection groups (no pathogen, one viral infection, one bacterial infection and co-infection) were analysed using Fisher's exact test for categorical data and the Kruskal-Wallis test for continuous data. Differences of proportions of resistant *Enterobacteriaceae* colonies between days 1 and 8, and days 8 and week 4 were assessed using the Wilcoxon matched pairs signed-rank test. In order to assess concordance of antibiotic use as determined by parent interviews and urine tests, a Kappa measure of agreement was used. All statistical tests were performed as two-tailed tests at 5% significance levels. SPSS version 20 was used for all analyses.

## Results

### Demographic and clinical data

Between February 2009 and February 2010 563 patients were enrolled, 545 returned for a follow-up visit at day 8, 13 were followed-up by telephone and 5 were lost to follow-up. Fig 1 shows a flowchart of enrolment and follow-up. Demographics, clinical diagnoses and outcome are displayed in Table 1. Patients had a median age of 1.96 years (IQR 1.05–3.18) and 95.2% of patients were aged 5 years or below. The male: female ratio was 1.28:1. Complete or partial recovery at day 8 according to the attending physician was reported for 138/563 (24.5%) and

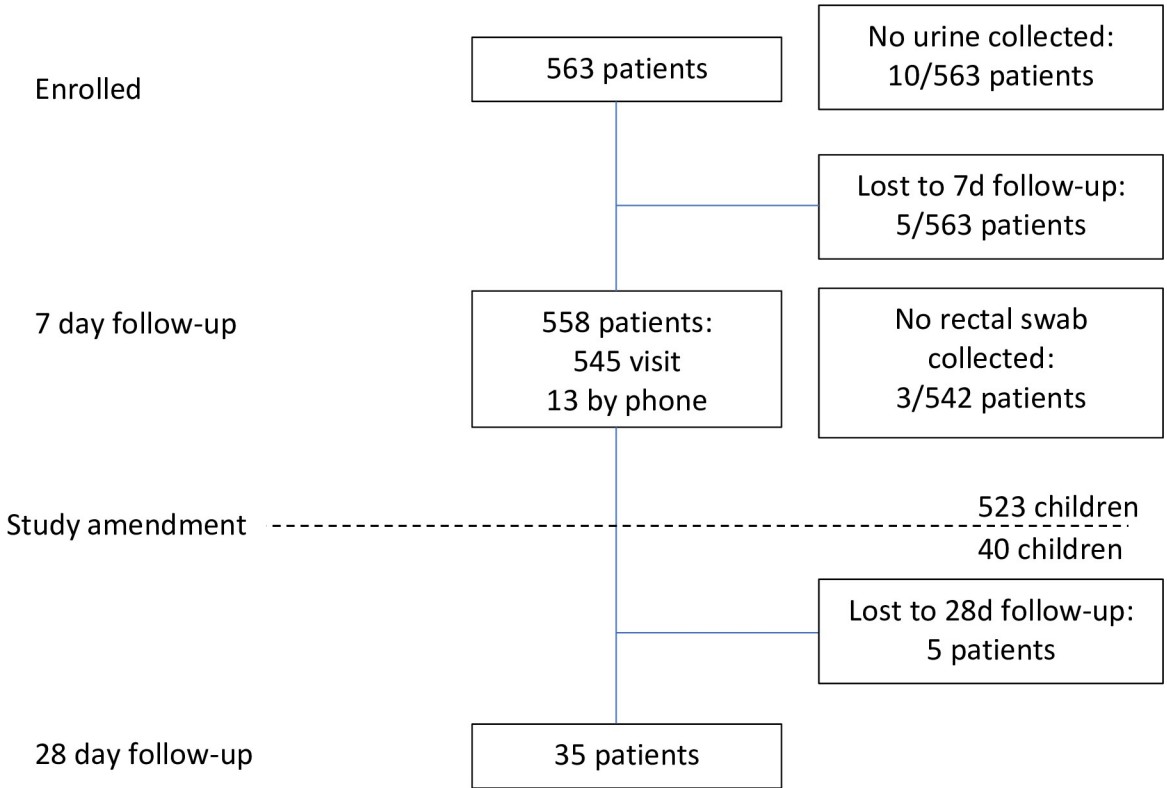

**Fig 1. Enrolment flowchart.** Flowchart of enrolment and follow-up of 563 patients with acute respiratory infection enrolled at the outpatient department of Children's Hospital 1, Ho Chi Minh City, Vietnam.

361/563 (64.1%) patients, respectively. In contrast, 59/563 (10.5%) patients did not recover: 6.4% remained clinically unchanged, 2.8% worsened and 1.2% were admitted to hospital. By proportion of fully recovered patients at day 8, upper respiratory infections had a better outcome than lower respiratory tract infections (34% [53/155] vs. 21% [81/391]) and, similarly, bronchitis had a worse outcome than other clinical diagnoses (20% [53/262]).

## Prescription of antibiotics and other medications

Antibiotics were prescribed for all but two patients (561/563, 99.6%), with a median duration of 6 days (IQR: 4–7). The most commonly prescribed antibiotics at presentation were amoxicillin-clavulanic acid (45.6%), cefuroxime axetil (22.0%), cefixime (11.4%), cefaclor (8.2%), erythromycin (3.7%), amoxicillin (3.0%), and cefpodoxime (2.3%). First choice antibiotics for ambulatory pneumonia treatment as recommended by the 2009 hospital guidelines, i.e. amoxicillin or co-trimoxazole, were infrequently prescribed (3.0 and 0.4%, respectively).

In addition to antibiotics, cough syrup (394/563, 70.0%), bronchodilators (324/563, 57.6%), antipyretics (150/563, 26.6%), antihistamines (66/563, 11.7%), mucolytics (62/563, 11.0%) and steroids (58/563, 10.3%) were also prescribed.

## Pathogen detection rates and associations

Detection of viral and bacterial pathogens using molecular assays is shown in Table 2. *S. pneumoniae* and *H. influenzae* type b were detected in 553/563 (98.4%) and 69/563 (12.3%) NTS samples from patients (Table 2). Results for these two bacteria were not included in further analyses (see discussion).

**Table 1. Patient baseline, diagnosis and outcome data.**

| Characteristics[a] | ARI patients (n = 563) |
|---|---|
| Median age in years (IQR) | 1.96 (1.05–3.18) |
| ≤ 1 year, n (%) | 137 (24.3) |
| 1-≤ 2 years, n (%) | 148 (26.3) |
| 2- ≤ 5 years, n (%) | 251 (44.6) |
| >5 years, n (%) | 27 (4.8) |
| Male, n (%) | 316 (56.1) |
| Clinical diagnosis[b] | |
| asthma | 17 (3.0) |
| bronchitis | 262 (46.5) |
| bronchiolitis | 122 (21.7) |
| pneumonia | 7 (1.2) |
| nasopharyngitis | 105 (18.7) |
| pharyngitis | 42 (7.5) |
| tonsillitis | 7 (1.2) |
| laryngotracheobronchitis | 1 (0.2) |
| Outcome[c] | |
| Complete recovery | 138 (24.5) |
| Partial recovery | 361 (64.1) |
| Unchanged | 36 (6.4) |
| Worsened | 16 (2.8) |
| Admitted | 7 (1.2) |
| Unknown | 5 (0.9) |

[a] Baseline characteristics of 563 patients with acute respiratory infection enrolled at the outpatient department of Children's Hospital 1, Ho Chi Minh City, Vietnam

[b] Clinical diagnosis of 563 patients with acute respiratory infection enrolled at the outpatient department of Children's Hospital 1, Ho Chi Minh City, Vietnam

[c] Outcome recorded at day 8 of 563 patients with acute respiratory infection enrolled at the outpatient department of Children's Hospital 1, Ho Chi Minh City, Vietnam.

Viruses were detected in 410/563 (72.8%) NTS samples, 294/410 (71.7%) were single infections and 116/410 (28.3%) were co-infections with either viruses or bacteria. Atypical bacteria and *Bordetella spp*. were detected in 43/563 (7.6%) samples, in most of which (27/43, 63%) viruses were also detected. Most commonly detected viruses were rhinoviruses (27%, 152/563) and enteroviruses (10.5%, 95/563). *M. pneumoniae* was the most frequently detected atypical bacterium (4.4%, 25/563).

Among patients older than five, a pathogen was detected in only 40% (11/27), as opposed to 80–90% in the <1, 1 and 2–5 age groups.

Detections by month of viral and atypical bacterial pathogens are shown in S1 Fig. Detection rates of pathogens varied from month to month during the year and clear seasonal patterns were observed for several viruses. Influenza virus A, Respiratory Syncytial Virus (RSV) and human metapneumovirus (hMPV) were mostly seen during and around the rainy season (June to November) with hMPV peaking after RSV, Parainfluenzavirus 3 (PIV3) was mostly detected from November to June. Enteroviruses and Rhinoviruses were detected throughout the year, with highest detection rates for Enteroviruses in November and for Rhinoviruses in March.

Significant associations between specific pathogens and clinical diagnosis or outcome were not observed, but co-infections and infections with multiple viruses were associated with a worse outcome.

**Table 2. Pathogens detected in respiratory samples.**

| | ARI patients |
|---|---|
| | (n = 563) |
| Viruses | |
| Influenza virus A, n(%) | 39 (6.9) |
| Influenza virus B, n(%) | 5 (0.9) |
| Enterovirus A-D, n(%) | 59 (10.5) |
| Adenovirus, n(%) | 52 (9.2) |
| Rhinovirus A-C, n(%) | 152 (27.0) |
| RSV A/B, n(%) | 54 (9.6) |
| Human metapneumovirus, n(%) | 41 (7.3) |
| Parainfluenza virus 1, n(%) | 10 (1.8) |
| Parainfluenza virus 2, n(%) | 11 (2.0) |
| Parainfluenza virus 3, n(%) | 46 (8.2) |
| Parainfluenza virus 4, n(%) | 19 (3.4) |
| Coronaviruses, n(%) | 21 (3.7) |
| Human parechovirus, n(%) | 4 (0.7) |
| Human bocavirus, n(%) | 19 (3.4) |
| Bacteria | |
| *Haemophilus influenzae*, n(%) | 69 (12.3) |
| *Streptococcus pneumoniae*, n(%) | 553 (98.4) |
| *Bordetella pertussis*, n(%) | 4 (0.7) |
| *Bordetella parapertussis*, n(%) | 11 (2.0) |
| *Legionella pneumophila*, n(%) | 0 (0) |
| *Mycoplasma pneumoniae*, n(%) | 25 (4.4) |
| *Chlamydophila pneumoniae*, n(%) | 3 (0.5) |
| *Chlamydophila psitacci*, n(%) | 0 (0) |
| Any pathogen positive, n(%) | 426 (75.6) |
| Single viral infection, n(%) | 294 (52.2) |
| Single bacterial infection, n(%) | 16 (2.8) |
| Co infection, n(%) | 116 (20.6) |
| No pathogen, n(%) | 137 (24.3) |

Viral and bacterial pathogens detected by real-time multiplex or single (RT-)PCR in pooled nasal and pharyngeal swabs taken at enrolment among 563 patients with acute respiratory infection enrolled at the outpatient department of Children's Hospital 1, Ho Chi Minh City, Vietnam.

Among the detected viruses, several are known to also be frequently detected among asymptomatic children, whereas influenza virus A and B, RSV, hMPV and PIV3 have stronger associations with disease and outbreaks of respiratory infections. These 4 were detected in 186/563 (33.0%) patients (influenza A 39, B 5, RSV 54, hMPV 41, PIV3 46, RSV + hMPV 1).

## Appropriateness of antibiotics according to guidelines and pathogens

We retrospectively classified antibiotic use as inappropriate based on clinical diagnosis and (RT-)PCR results. We used two definitions for inappropriate antibiotic use: 1) based on clinical diagnosis, *i.e.* antibiotic use in patients who were not diagnosed with pneumonia or pharyngitis/tonsillitis and 2) based on a combination of clinical and laboratory diagnosis in patients whose respiratory samples were positive for at least one of 14 viruses and negative for 6 bacterial pathogens tested by PCR (excluding *S. pneumoniae* and *H. influenzae*).

**Table 3. Antibiotic use assessed by questionnaire and HPLC.**

| | HPLC | (%) | Questionnaire | (%) | of which HPLC positive |
|---|---|---|---|---|---|
| | (total number = 553) | | (total number = 553) | | |
| Any antibiotic (patients) | 178 | 32.2 | 123 | 22.2 | 60 |
| 1 antibiotic | 160 | 28.9 | 119 | 21.5 | |
| 2 antibiotics | 14 | 2.5 | 4 | 0.7 | |
| 3 antibiotics | 4 | 0.7 | | | |
| Total (antibiotics) | 200 | | 127 | | |
| Per antibioitc | | | | | |
| Amoxicillin (with or without clavulanic acid) | 52 | 9.4 | 48 | 8.7 | 22 |
| Cefaclor | 28 | 5.1 | 37 | 6.7 | 12 |
| Cefadroxil | 34 | 6.1 | 3 | 0.5 | 3 |
| Cefixime | 54 | 9.8 | 14 | 2.5 | 8 |
| Cefuroxime | 10 | 1.8 | 16 | 2.9 | 4 |
| Cephalexin | 22 | 4.0 | 1 | 0.2 | 1 |
| Other | | | 8 | 0.2 | |
| Total (antibiotics) | 200 | | 127 | | 60 |
| Negative (below the level of detection) | 375 | 67.8 | | | |
| Denied use | | | 206 | 37.3 | 29 |
| Unknown | | | 224 | 40.5 | 89 |

Antibiotic use among 553 patients with acute respiratory infection prior to enrolment at the outpatient department of Children's Hospital 1, Ho Chi Minh City, Vietnam as assessed by questionnaire on enrolment and by HPLC of urine collected on enrolment.

Only 7 (1.2%) patients had a clinical diagnosis of pneumonia and 49 (8.7%) patients had a diagnosis of pharyngitis or tonsillitis (Table 1). The remaining ninety percent (90.1%, 507/563) of patients received antibiotics inappropriately according to definition 1. First choice antibiotics for ambulatory pneumonia treatment as recommended by the 2009 hospital guidelines, i.e. amoxicillin or co-trimoxazole, were rarely prescribed (3.0 and 0.4%, respectively). Among these 507 patients, 380 had a virus and no atypical bacteria detected, and therefore, according to definition 2, 67.5% (380/563) of patients received antibiotics inappropriately.

## Antibiotic use on presentation by questionnaire and by detection in urine by HPLC

Results from parent interviews on antibiotic use and HPLC detection in urine collected at presentation are shown in Tables 3 and 4. From 553/563 (98.2%) patients a urine sample was collected at presentation. Antibiotic use during the two days prior to enrolment was reported by 123/553 (22.2%) of parents, 206/553 (37%) reported no antibiotic use and 224/553 (40.5%) did

**Table 4. Kappa score.**

| Kappa | 0.37 | | questionnaire | | |
|---|---|---|---|---|---|
| | | | + | - | |
| HPLC | + | 60 | 29 | 89 |
| | - | 63 | 177 | 240 |
| | | 123 | 206 | 329 |

Kappa score of antibiotic use as assessed by questionnaire on enrolment and by HPLC of urine collected on enrolment.

not know. Among patients from whom a urine sample was collected, 32% (178/553) had a positive result for one of 6 frequently prescribed antibiotics detected by HPLC in urine; 2 different antibiotics were detected in 14 patients and 3 in 4 patients (Table 3). Cefixime, a third-generation oral cephalosporin, was the most frequently detected antibiotic.

Among those who confirmed or denied antibiotic use, there was agreement with HPLC results in 237/329 (72%; kappa score 0.37 [0.21–0.40: fair agreement], Table 4). Antibiotics were detected in urine of 29/206 (14%) patients whose parents denied use, and in 60/123 (49%) patients for whom use was reported, with the measured antibiotic corresponding to the reported antibiotic in 50/60 (83%) of cases. Use of the 6 tested antibiotics was reported in 56 of the remaining 63 (89%) cases that tested negative. In patients for whom prior use was unknown, antibiotics were detected in 89/224 (40%). The sensitivity of HPLC to detect 6 antibiotics in urine from patients whose parents reported use of these same 6 antibiotics, was 49%.

## Effect of antibiotics on intestinal *Enterobacteriaceae*

We determined the proportion of patients carrying resistant *Enterobacteriaceae* and the fraction of antibiotic resistant intestinal *Enterobacteriaceae* by semi-quantitative culture of rectal swabs taken on day 1 (n = 563) and 8 (n = 542) on 9 MacConkey agar plates each: with and different antibiotics (tetracycline, amoxicillin, amoxicillin-clavulanic acid, ceftazidime, ciprofloxacin, trimethoprim/sulfamethoxazole, gentamicin, meropenem). Thirty-five of 40 consecutive invited patients came back to hospital after 4 weeks for re-assessment of intestinal *Enterobacteriaceae*, after interim analysis of the first 100 patients had shown strong increments of the proportion of patients carrying resistant bacteria and the proportions of resistant bacteria in their samples (Fig 2). Proportions of patients and proportions of counted resistant colonies are shown in Table 5. McNemar test and Wilcoxon matched pairs signed-rank tests were used to assess statistical significance of different proportions of patients and resistant colonies, respectively. Log transformed colony counts at 3 different timepoints on plates with and without 4 different antibiotics (amoxicillin, ceftazidime, gentamicin, and ciprofloxacin) are shown in Fig 2. Regression was used to calculate the position of lines representing the mean.

The proportion of patients from whom antibiotic resistant *Enterobacteriaceae* were cultured on 8 plates containing different antibiotics and the proportion of counted resistant *Enterobacteriaceae* colonies (defined as the ratios of the number of colonies on 8 different plates containing 8 different antibiotics divided by the number of counted colonies on 1 plate without antibiotics) among 563, 542 and 35 patients with acute respiratory infection enrolled at the outpatient department of Children's Hospital 1, Ho Chi Minh City, Vietnam at day 1, day and day 28 of enrolment. McNemar test and Wilcoxon matched pairs signed-rank tests were used to assess statistical significance of different proportions of patients and resistant colonies, respectively.

For most antibiotics, the proportions of patients carrying resistant bacteria were already high at day 1. These proportions further increased significantly at day 8 for amoxicillin, amoxicillin with clavulanic acid, ceftazidime, ciprofloxacin and gentamicin, and restored to baseline proportions or below after 4 weeks. Similarly, the proportions of resistant intestinal *Enterobacteriaceae* colonies increased for these 5 antibiotics and returned to day 1 levels after 4 weeks (Table 5 and Fig 2). For ceftazidime and co-trimoxazole the proportions of resistant *Enterobacteriaceae* colonies were significantly lower after 4 weeks compared to day 1, which may be explained by antibiotic use before presentation. Indeed, when excluding patients who were already using antibiotics on day 1 (assessed by questionnaire or HPLC), proportions of resistant *Enterobacteriaceae* colonies after 4 weeks were similar to those at day 1.

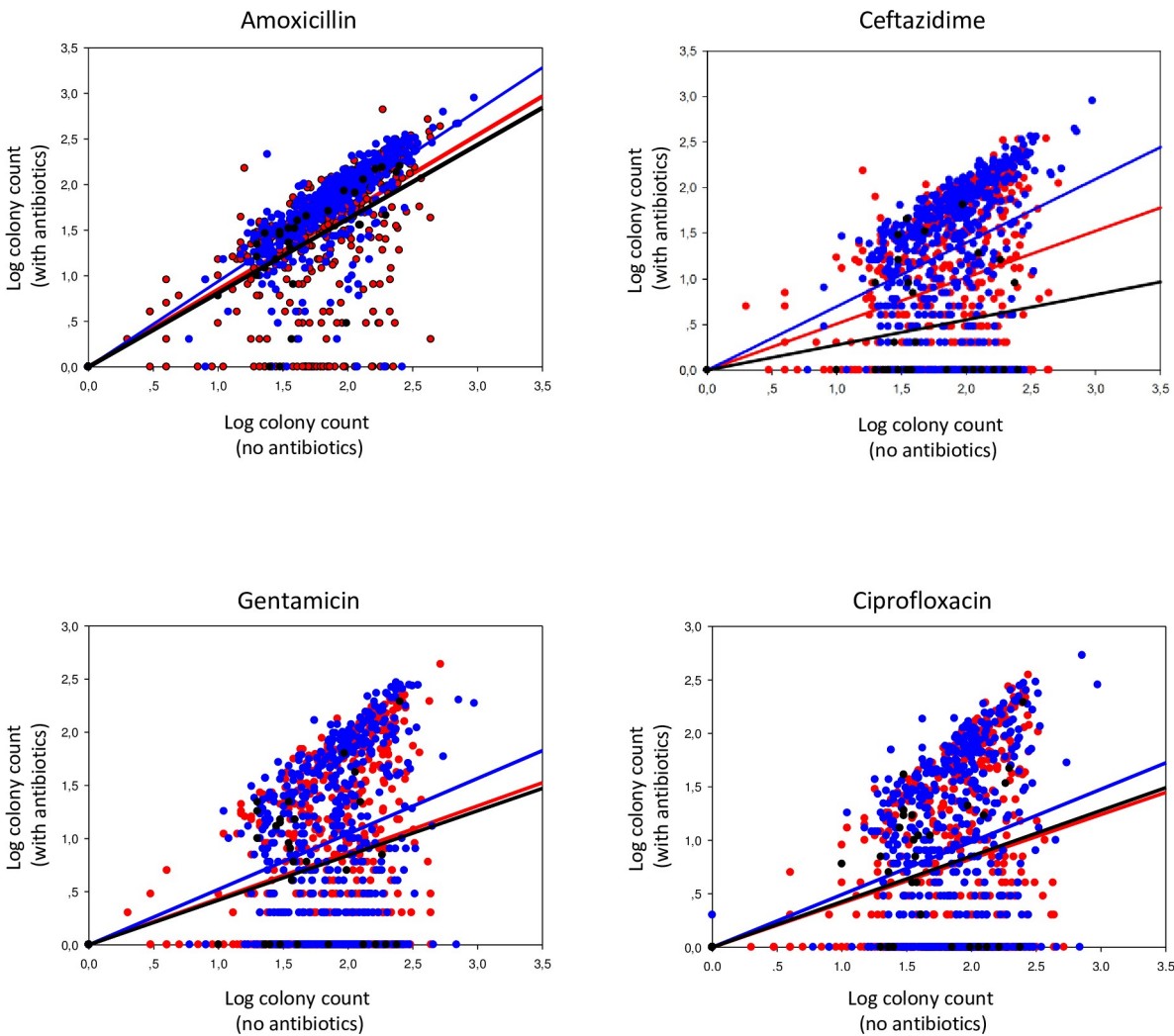

**Fig 2. Gut *Enterobacteriaceae* grown on plates with and without antibiotics.** Logarithmically transformed colony counts of *Enterobacteriaceae* on MacConkey agar with (y-axis) and without (x-axis, MC) antibiotics from rectal swabs taken on day 1 (n = 563, blue), day 8 (n = 542, red) and day 29 (n = 35, black) from patients with acute respiratory infection enrolled at the outpatient department of Children's Hospital 1, Ho Chi Minh City, Vietnam. Red, blue and black lines represent the mean as determined by regression analysis.

## Discussion

According to systematic reviews and current international and local guidelines, including those in place at the recruiting hospital, only clinical diagnoses of pneumonia and streptococcal pharyngitis/tonsillitis are indications for prescription of antibiotics in outpatients with ARI [14–16, 18–20]. Antibiotics have not been proven effective and therefore are not recommended for other diagnoses unless there are bacterial super-infections, but these have been reported to occur in only around 5–8% of children with ARI [32, 33]. In our study, pneumonia and pharyngitis/tonsillitis were diagnosed in only 1.2% (n = 7) and 8.7% (n = 49) of patients presenting with ARI, but antibiotics were prescribed in all but two patients enrolled. This means that more than 90% of patients inappropriately received antibiotics for clinical diagnoses, which do not require antibiotic treatment according to treatment guidelines. Using more stringent criteria, based on detection of viral and bacterial pathogens (although tested retrospectively in this study), nearly 70% of patients were inappropriately treated with antibiotics, *i*.

**Table 5. Proportions of patients with resistant *Enterobacteriaceae* and proportions of resistant *Enterobacteriaceae* at different timepoints.**

| Proportion of children carrying resistant bacteria | Day 1 (n = 563) | Day 8 (n = 542) | Day 28 (n = 35) | p1 (Day 1 vs 8) | p2 (Day 1 vs 28) |
|---|---|---|---|---|---|
| Amoxicillin, n (%) | 513 (91.1) | 516 (95.2) | 28 (80.0) | 0.008 | 0.5 |
| Amoxicillin—Clavulanic acid, n (%) | 504 (89.5) | 513 (94.7) | 28 (80.0) | 0.001 | 1 |
| Ceftazidime, n (%) | 379 (67.3) | 446 (82.3) | 13 (37.1) | <0.001 | 0.2 |
| Ciprofloxacin, n (%) | 322 (57.2) | 361 (66.6) | 20 (57.1) | <0.001 | 0.2 |
| Gentamicin, n (%) | 333 (59.1) | 377 (69.6) | 19 (54.3) | <0.001 | 0.3 |
| Tetracycline, n (%) | 516 (91.7) | 498 (91.9) | 29 (82.9) | 1 | 0.5 |
| Cotrimoxazole, n (%) | 524 (93.1) | 513 (94.6) | 25 (71.4) | 0.4 | 0.3 |
| Meropenem, n (%) | 4 (0.7) | 2 (0.4) | 0 (0) | 0.7 | NA |
| Proportion of resistant bacteria | | | | | |
| Amoxicillin, n (%) | 0.84 | 0.94 | 0.79 | <0.001 | 0.06 |
| Amoxicillin—Clavulanic acid, n (%) | 0.82 | 0.92 | 0.82 | <0.001 | 0.87 |
| Ceftazidime, n (%) | 0.50 | 0.70 | 0.29 | <0.001 | 0.005 |
| Ciprofloxacin, n (%) | 0.40 | 0.50 | 0.44 | <0.001 | 0.1 |
| Gentamicin, n (%) | 0.41 | 0.52 | 0.44 | <0.001 | 0.7 |
| Tetracycline, n (%) | 0.82 | 0.84 | 0.83 | 0.36 | 0.28 |
| Cotrimoxazole, n (%) | 0.85 | 0.89 | 0.68 | 0.054 | 0.02 |
| Meropenem, n (%) | 0.004 | 0.002 | 0 | 0.5 | 0.3 |

*e.* when only a viral pathogen was detected. Moreover, the choice of antibiotics rarely followed hospital guidelines, but instead broad-spectrum agents, such as amoxicillin-clavulanic acid and second or third generation cephalosporins were prescribed which are not recommended for ARIs.

In addition to high rates of inappropriate antibiotic prescription by attending physicians, antibiotics were also frequently used prior to presentation. Interviewed parents reported antibiotic use before presentation in 22% of children, while 41% of parents did not know whether antibiotics were used. As determined by HPLC detection of antibiotics in urine, the rate of prior use was 32% with fair agreement between reported and detected antibiotics (Table 4). HPLC may provide a more accurate and objective detection rate of antibiotic use than the questionnaire, although sensitivity is limited and HPLC only detects antibiotics several hours after use. Antibiotics were detected by HPLC in only 49% of samples from patients whose parents reported antibiotic use. If we extrapolate this, we can then also assume that the antibiotic use among patients whose parents didn't report or weren't sure about antibiotic use is in reality also approximately twice as high as detected with HPLC: 28% and 80%, respectively, or a total of 65% of patients who had used antibiotics before presentation. These high reported, measured and estimated rates of antibiotic use reflect the over-the-counter availability and widespread use of antibiotics in Vietnam.

High rates of antibiotic use in outpatient children with ARI have been reported before, particularly in Asian countries, with rates ranging from 30% in the UK to around 80% in China and Korea [34–36]. Although differences may be partially explained by the use of different definitions and differences in study populations, they support our observed high level of antibiotic overuse in Vietnamese children. A high rate of antibiotic consumption in Vietnamese children was also suggested by a community survey in northern Vietnam where 73% of interviewed parents indicated that their children between 1–5 years old had used antibiotics in the preceding month [13].

That high usage of antibiotics is not without consequences was shown by the selection of antibiotic-resistant *Enterobacteriaceae* in stool samples of our patients. The proportions of

patients carrying resistant bacteria as well as the proportions of resistant *Enterobacteriaceae* colonies in stools of individual patients to commonly used antibiotics were already high at presentation but showed further significant increases at day 8. This included *Enterobacteriaceae* resistant to aminoglycosides or quinolones, representing drugs that were not frequently prescribed for ARI, presumably reflecting the presence of multiple resistance genes on mobile genetic elements such as plasmids. The significant increase in resistant bacteria under antibiotic pressure may lead to increased transfer of resistance genes to other commensal or pathogenic bacteria, and to person-to-person spread in the community through faecal-oral transmission. After withdrawing antibiotics, we observed restoration to baseline values of proportions of resistant *Enterobacteriaceae* colonies for all antibiotics tested. Proportions of *Enterobacteriaceae* colonies resistant to ceftazidime and co-trimoxazole were restored to below baseline values, which may be explained by antibiotic use before presentation. While these observations suggest that immediate effects of antibiotic treatment on selection of resistant bacteria are only short-lived, long-term persistence of selected resistance genes has been shown in other studies [37–42]. Given the widespread use of antibiotics in Vietnam, the selection, persistence and transmission of resistant bacteria likely occurs on a large scale at community levels, as is also indicated by the high prevalence of resistance at presentation in our outpatients.

We acknowledge there were a number of limitations to this work, *e.g.* clinical diagnoses were not assessed and recorded in a systematic and standardized way. Although we acknowledge this may be seen as a limitation, our work reflects the real life day-to-day practice in the busy outpatient clinics of a paediatric hospital and representative high antibiotic prescription rates. Results presented here may not be generalisable to countries where there is no over-the-counter availability of antibiotics. Other limitations may have been:

1. Attending physicians recorded clinical diagnoses at their discretion without the use of exact case definitions. Theoretically, doctors may have over-diagnosed conditions as pneumonia to justify antibiotic prescriptions, although this did not seem to have occurred.

2. We did not record any laboratory values that are markers of bacterial infection like white blood cell count and C-reactive protein. This is not routinely done in the outpatient clinics where patients were recruited and results would have influenced the observational nature of this study.

3. The CRF we used did not specifically record (warning signs of) complications as mentioned in some guidelines, *e.g.* NICE criteria for complications or Centor criteria for streptococcal pharyngitis [43, 44].

4. We used the respiratory examination rooms for recruitment where the relatively more severe cases of respiratory infections are seen. This may have selected for a patient population more likely to receive antibiotics.

5. Common childhood bacterial pneumonia pathogens *S. pneumoniae* and *H. influenzae* type b were not optimally diagnosed in this study. Respiratory infection with these pathogens should be assessed using wet and Gram-staining and bacterial culture of sputum, which was not performed here. Molecular testing results were not taken into account when describing aetiology as these results were considered to include colonisation, and in the case of *S. pneumoniae* potentially cross-reactivity. High carriage rates of *S. pneumoniae* up to 80% have been reported in children in the region, but we didn't find report of rates as high as found in our cohort [45, 46]. Our PCR assay for *S. pneumoniae* targeted the pneumolysin gene [21]. This gene is a virulence factor and expressed by almost all isolates of *S.*

*pneumoniae.* However, recent reports also showed that other species of *Streptococcus* can also express this gene [47, 48]. Our assay was developed for use in diagnosis of central nervous system infections in otherwise sterile cerebrospinal fluid. It is not unlikely that when used in respiratory swabs cross-reactivity with other oral streptococci may occur. We further justified disregarding these results because we detected these bacteria at similar rates in NTS samples from a cohort of healthy children (unpublished data). The bacterial load, as expressed by Cp value for *S. pneumoniae* was significantly higher in patients than in healthy children, however, this did not correlate with disease, disease severity, or whether other pathogens were detected. We hypothesized that this higher load, which has also been observed in other studies, was caused by increased secretions and shedding of (colonized) nasopharyngeal epithelial cells and results for these two bacteria were not included in this analysis [49].

We show that in a setting of unrestricted access to antibiotics and non-adherence to clinical guidelines there is an alarmingly high inappropriate use of antibiotics for children with mild self-limiting illness, before visiting the outpatient clinic and prescribed by doctors in the outpatient clinic. This situation is similar in many low- and middle-income countries across Asia and Africa [50]. We also show that taking antibiotics exerts strong but reversible selective pressure on intestinal *Enterobacteriaeceae.*

Physicians report that they prescribe antibiotics because of a lack of diagnostic tools to help discriminate between bacterial and viral infections. Laboratory capacity building, including classical microbiology and molecular testing for viruses could help provide this information for more severe patients and inpatients within hours to days, which is too slow for outpatients. Point-of-care testing for 4 common viruses associated with ARI (Influenza virus A and B, RSV, hMPV and PIV-3) might have prevented antibiotic prescription in 186/563 (33%) of our patients, assuming good sensitivity and adherence. Likewise, use of a rapid CRP test could help distinguish viral from bacterial infections and reduce antibiotic prescription rates significantly, as was recently shown in a study in northern Vietnam [51].

In 2013, Vietnam established a National Action Plan for AMR of which, similar to the WHO Global Action Plan, key components are to raise public awareness, to strengthen and improve national surveillance on antibiotic use and susceptibility, to ensure adequate supply of quality antibiotics, to promote the proper and safe use of antibiotics in humans, livestock, poultry and aquaculture and to promote infection control [52]. Based on our observations, and more recent surveillance studies on antimicrobial resistance in Vietnam, we propose that Vietnam's strategy for AMR should, besides enhancement of laboratory support, include a focus on intensive and continued medical training (CME) in rational antibiotic use for medical students and certified medical staff including physicians and development and implementation of antibiotic stewardship programmes and treatment guidelines at all levels of the healthcare system, similar to but more extensive than what published in a recent study [53]. It is crucial to involve commercial pharmacies in this plan, as 80% of antibiotics for human use are dispensed in the 60.000 pharmacies in Vietnam (of which 90% without prescription) and these are dependent on antibiotic sales for 25% of their revenue [12]. The general Vietnamese public should be engaged through educational programmes through national media and in the communities. At government level, existing legislation on prescription-only drugs should be enforced and expanded to assure that antibiotics can no longer be purchased without a doctor's consultation.

## Supporting information

**S1 Fig. Pathogen detection by month.** Detections by month of viral and atypical bacterial pathogens detected by real-time multiplex or single (RT-)PCR in pooled nasal and pharyngeal

swabs taken at enrolment among 563 patients with acute respiratory infection enrolled at the outpatient department of Children's Hospital 1, Ho Chi Minh City, Vietnam. X-axis: month; Y-axis: number of positive cases per pathogen. FluA: Influenza virus A; FluB: Influenza virus B; RSV A/B: Respiratory Syncytial Virus A and B; PIV1-4: Human parainfluenza viruses 1–4; hRV: Human Rhinovirus; EV: Enterovirus A-D; CoV: Human Coronavirus; BoV: Human Bocavirus; MPV: Human Metapneumovirus; PeV: Human Parechovirus; AdV: Adenovirus; MP: *Mycoplasma pneumoniae;* CPn: *Chlamydophila pneumoniae;* CPs: *Chlamydophila psitacci;* LP: *Legionella pneumophila;* BPt: *Bordetella pertussis;* BPp: *Bordetella parapertussis.*
(TIF)

**S1 Table. Prescribed antibiotics.** (a) List of most frequently prescribed antibiotics in the outpatient department in 2007 of Children's Hospital 1, Ho Chi Minh City, Vietnam, (b) List of most frequently sold antibiotics for outpatient use in 2007 in the hospital pharmacy of Children's Hospital 1, Ho Chi Minh City, Vietnam.
(XLSX)

**S1 Raw data.**
(XLSX)

## Author Contributions

**Conceptualization:** Ngo Ngoc Quang Minh, Thomas Pouplin, Stephen Baker, Constance Schultsz, Menno D. de Jong, H. Rogier van Doorn.

**Formal analysis:** Ngo Ngoc Quang Minh, Pham Van Toi, Nguyen Hanh Uyen, Vu Thi Ty Hang, Nguyen Thi Thuy Chinh B'Krong, Nguyen Thi Tham, Thai Dang Khoa, Huynh Duy Khuong, Pham Quynh Vi, Nguyen Ngoc Hong Phuc, Le Thi Minh Vien, Doan Van Khanh, Pham Nguyen Phuong, Phung Khanh Lam, H. Rogier van Doorn.

**Funding acquisition:** Menno D. de Jong, H. Rogier van Doorn.

**Investigation:** Ngo Ngoc Quang Minh.

**Methodology:** Ngo Ngoc Quang Minh, Pham Van Toi, Nguyen Hanh Uyen, Vu Thi Ty Hang, Nguyen Thi Thuy Chinh B'Krong, Nguyen Thi Tham, Thai Dang Khoa, Huynh Duy Khuong, Nguyen Ngoc Hong Phuc, Le Thi Minh Vien, Thomas Pouplin, Doan Van Khanh, Pham Nguyen Phuong, Phung Khanh Lam, Heiman F. L. Wertheim, James I. Campbell, Stephen Baker, Constance Schultsz, Menno D. de Jong, H. Rogier van Doorn.

**Project administration:** Ngo Ngoc Quang Minh.

**Supervision:** Ngo Ngoc Quang Minh, Pham Van Toi, Le Minh Qui, Le Binh Bao Tinh, Nguyen Thi Ngoc, Le Thi Ngoc Kim, Vu Thi Ty Hang, Thai Dang Khoa, Thomas Pouplin, Heiman F. L. Wertheim, James I. Campbell, Christopher M. Parry, Juliet E. Bryant, Constance Schultsz, Nguyen Thanh Hung, Menno D. de Jong, H. Rogier van Doorn.

**Validation:** Nguyen Hanh Uyen, Vu Thi Ty Hang, Thomas Pouplin, H. Rogier van Doorn.

**Writing – original draft:** Ngo Ngoc Quang Minh, H. Rogier van Doorn.

**Writing – review & editing:** Ngo Ngoc Quang Minh, Pham Van Toi, Le Minh Qui, Le Binh Bao Tinh, Nguyen Thi Ngoc, Le Thi Ngoc Kim, Nguyen Hanh Uyen, Vu Thi Ty Hang, Nguyen Thi Thuy Chinh B'Krong, Nguyen Thi Tham, Thai Dang Khoa, Huynh Duy Khuong, Pham Quynh Vi, Nguyen Ngoc Hong Phuc, Le Thi Minh Vien, Thomas Pouplin, Doan Van Khanh, Pham Nguyen Phuong, Phung Khanh Lam, Heiman F. L. Wertheim,

James I. Campbell, Stephen Baker, Christopher M. Parry, Juliet E. Bryant, Constance Schultsz, Nguyen Thanh Hung, Menno D. de Jong, H. Rogier van Doorn.

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
