## [Decision Letter · Decision Letter 0]

17 Jan 2020

PONE-D-19-28321

Antibiotic use and prescription and its effects on intestinal flora in children with mild respiratory infections in Ho Chi Minh City, Vietnam. A prospective observational outpatient study.

PLOS ONE

Dear Dr. van Doorn,

Thank you for submitting your manuscript to PLOS ONE. After careful consideration, we have decided that your manuscript does not meet our criteria for publication and must therefore be rejected.

I am sorry that we cannot be more positive on this occasion, but hope that you appreciate the reasons for this decision.

Yours sincerely,

Mehreen Arshad, M.D.

Academic Editor

PLOS ONE

Additional Editor Comments (if provided):

The authors of this manuscript attempt to determine the overuse of antibiotics in children with URI. While the premise of this is interesting this study does not add new information to this research field. The objective of this study is also unclear. They have looked at resistant organisms very superficially. The 28 day follow up is only done on 35 kids without an explanation on why such few children were enrolled and how they were chosen. The authors have made several comments without showing any supporting data. No details are given on how the molecular diagnostics were done. Specific comments are below:

Introduction

Line 38: data from UK seems out of place, would recommend using Asian countries or those of similar economic status

Methods:

Line 62: how was the follow up subset decided upon? Was there a selection criterion of some sort?

Line 66: was every patient enrolled in that 1 hr? Were they randomized in any way?

Line 89: Was the testing done a standard of care? Is this a commercial assay?

Results:

Line 126: This should be in a flow chart

Line 127-129: This sentence should either be deleted or the data shown.

Line 146-148: It is hard to believe that almost all children were colonized with S. pneumoniae, which then brings in to question the validity of the entire assay. No control data is shown either.

Line 168: It is unclear why the urine HPLC was done. It would have made more sense if the authors stated the objectives clearly in the beginning.

The discussion is too wordy, and comes across as a policy paper in some places.

Figure 1: the x and y axis does not make sense. What is this a log of? CFU/ml? What does MC stand for?

There is no discussion of the supplementary figure or table in the main text.

Reviewers' comments:

Reviewer's Responses to Questions

**Comments to the Author**

1. Is the manuscript technically sound, and do the data support the conclusions?

Reviewer #1: No

2. Has the statistical analysis been performed appropriately and rigorously? 

Reviewer #1: No

3. Have the authors made all data underlying the findings in their manuscript fully available?

Reviewer #1: Yes

4. Is the manuscript presented in an intelligible fashion and written in standard English?

Reviewer #1: No

5. Review Comments to the Author

Reviewer #1: The manuscript prepared by Quang Minh et al presented a report on unnecessary usage of antibiotics in the Vietnamese population. The author summarized data collection on 563 children and evaluated the appropriateness of antibiotic use on presentation and proportion of resistant enterobacteriaceae in the gut flora before, during and after antibiotic prescription.

My major critique of the manuscript is the presentation and analyses of data do not align with the objectives of the manuscript. All objectives should be clearly mentioned in the introduction section. The table and figures are not explained well in the results section, which creates a problem for a reader to understand the objectives and actual summary of the results. I would suggest to rewrite the result and discussion section.

Regression analysis is not a suitable analysis for showing effect of antibiotics on normal flora. Authors has not discussed which colonies they picked and no details were given on bacterial identification. Data is not substantial to support the semi-quantitative quantification of normal flora.

Figure S1 is about the seasonality of pathogens , I am confused, there is no discussion about seasonality. It was discussed out of context in line # 151. Among pathogens detected Strep. Pneumonia (98%), H.influenzae (12.3%) and rhinovirus (27%), somehow the discussion on rhinovirus was left out.

Similarly, table 4 is hard to understand. The author should explain a little bit about the fraction of bacteria and antibiotics.

Add a sentence for a rationale of semi-quantitative detection and how well it represents normal intestinal flora? It was not mentioned whether antibiotics were added in media or it was measured through disk diffusion?

Molecular diagnostics: the details of methods should be given, with the name of the platform, principle of test and details of kits used.

Overall, this manuscript requires language editing as sentence structure is not correct at various places and grammatical mistakes should be corrected.

various sentences donot make sense for example:

line 180-182 212-213, 221-222

Specific comments:

Abstract:

Objective: line# 3-5 needs to be rephrased

The word moleculary should be changed to molecular detection

Line#38: The term GP stands for..?

Line #131-139: data is not shown in any table,

Line# 161: worse outcome should be defined if it is hospital discharge then analyses should be changed accordingly.

6. PLOS authors have the option to publish the peer review history of their article (what does this mean?). If published, this will include your full peer review and any attached files.

Reviewer #1: No

- - - - -

---

## [Author Response · Author response to Decision Letter 0]

18 Mar 2020

We have pasted the editor’s and reviewer’s comments and indicated our replies to them by “>”

The authors of this manuscript attempt to determine the overuse of antibiotics in children with URI. While the premise of this is interesting this study does not add new information to this research field. 

>We strongly believe that the systematic and comprehensive way in which we assessed several aspects of outpatient treatment of mild respiratory infections and the impact of antibiotic use adds to the field. While the high rates of antibiotic prescription for ARI in Vietnam are not new, assessing of pre-presentation antibiotic use and the comparison between reported and measured use (using HPLC in urine) certainly is in this setting. Furthermore, the semi-quantitative assessment of selection of resistant flora from longitudinal samples showing the negative impact of unnecessary antibiotic use using standard culture methods has to our knowledge not been done before and shows a very consistent and highly significant difference between resistant fractions among samples on presentation and after antibiotic use among >500 children against 7 out of 8 tested drugs, that was restored in a subset of patients after 28 days. 

The objective of this study is also unclear. 

>We have more explicitly clarified and expanded further on the objectives that were listed in abstract (line 5-9) and introduction (line 52-58)

They have looked at resistant organisms very superficially. 

>We have expanded on the methods (lines 122-137, 147-149), interpretation and analysis (lines 233-254 and legends Figure 2 and table 4) to make clearer what was done and that this was not superficial. Semi-quantitative culture was used to assess the proportions of resistant colonies of lactose fermenting Enterobacteriaceae to 8 different drugs by culturing rectal swabs on MacConkey agar with and without antibiotics. More than 1100 faecal specimens were cultured and counted on 9 plates each.

The 28 day follow up is only done on 35 kids without an explanation on why such few children were enrolled and how they were chosen. 

>This additional sample was collected from 35 of the last 40 patients who were enrolled into the study. This amendment to the protocol was made after a very strong selective effect was seen after interim analysis of the first 100 patients. Added to the manuscript in lines 70-73, 237-240)

The authors have made several comments without showing any supporting data. 

>The editor did not indicate the instances where we made comments without supporting data. We have gone through the manuscript and think we have addressed these. There are now two instances left where we say “data not shown” or “unpublished data”. One concerns a discussion section of relative loads of S. pneumoniae and H. influenzae in samples from our study patients vs healthy volunteers form an unpublished study. We do not think including these data will contribute to the discussion or the justification of why these data weren’t used in the analysis of our study. The second concerns the exact data of switching regimens during different visits to the outpatient clinic. Although we don’t show the interpretation and analysis of these data, they are included in the raw dataset we uploaded.

No details are given on how the molecular diagnostics were done. 

>References for the assays were given, we have published the use of these assays previously. We have added information from these previous publications and expanded on the described methods (line 92-110). 

Specific comments are below:

Introduction

Line 38: data from UK seems out of place, would recommend using Asian countries or those of similar economic status

>Added examples from Vietnam in lines 41-51

Methods:

Line 62: how was the follow up subset decided upon? Was there a selection criterion of some sort?

>See above

Line 66: was every patient enrolled in that 1 hr? Were they randomized in any way?

>No randomisation was done. One study doctor was present at the outpatient clinic during this hour and enrolled the first eligible patient from the waiting list, helped the clinic doctors with consent and CRF completion and then moved on to enrol the next eligible patient etc.

Line 89: Was the testing done as standard of care? Is this a commercial assay?

>No and no, added to the manuscript (lines 92-93, 109-110)

Results:

Line 126: This should be in a flow chart

>Added as Figure 1

Line 127-129: This sentence should either be deleted or the data shown.

>Data were added (lines 161-164)

Line 146-148: It is hard to believe that almost all children were colonized with S. pneumoniae, which then brings in to question the validity of the entire assay. No control data is shown either.

>We have not used these data for analysis and have added more specific comments / justification / explanation on this in the discussion (lines 325-344).

Line 168: It is unclear why the urine HPLC was done. It would have made more sense if the authors stated the objectives clearly in the beginning.

>This is now stated more clearly in abstract (line 5-9) and introduction (line 52-58)

The discussion is too wordy, and comes across as a policy paper in some places.

>We have reviewed the discussion. We do not think it is too wordy for a journal without word limit. We think the policy parts (assuming the editor means lines 358-373) contain relevant context and contextual recommendations for the readers 

Figure 1: the x and y axis does not make sense. What is this a log of? CFU/ml? What does MC stand for?

> MC stands for MacConkey agar, i.e. agar without antibiotics added, logs are of colony counts. This was stated in the legends. We have tried to show this clearer in both legends to figure 2 and methods (lines 135-137).

There is no discussion of the supplementary figure or table in the main text.

>Lines referring to this table and figure were moved / added to make their purpose and what is shown more clear (lines 55-58 and 189-194)

Reviewers' comments:

Reviewer's Responses to Questions

Comments to the Author

1. Is the manuscript technically sound, and do the data support the conclusions?

Reviewer #1: No

2. Has the statistical analysis been performed appropriately and rigorously?

Reviewer #1: No

3. Have the authors made all data underlying the findings in their manuscript fully available?

Reviewer #1: Yes

4. Is the manuscript presented in an intelligible fashion and written in standard English?

Reviewer #1: No

5. Review Comments to the Author

Reviewer #1: The manuscript prepared by Quang Minh et al presented a report on unnecessary usage of antibiotics in the Vietnamese population. The author summarized data collection on 563 children and evaluated the appropriateness of antibiotic use on presentation and proportion of resistant enterobacteriaceae in the gut flora before, during and after antibiotic prescription.

My major critique of the manuscript is the presentation and analyses of data do not align with the objectives of the manuscript. All objectives should be clearly mentioned in the introduction section. 

>Added in abstract (line 5-9) and introduction (line 52-58)

The table and figures are not explained well in the results section, which creates a problem for a reader to understand the objectives and actual summary of the results. I would suggest to rewrite the result and discussion section.

>We acknowledge this wasn’t done very well and has been changed, see also replies to editors comments. All figures are better referenced in the body of text, legends have been expanded where needed and parts of legends have been copied in the text for better reading and easier understanding.

Regression analysis is not a suitable analysis for showing effect of antibiotics on normal flora. 

>The reviewer is of the opinion that something was done incorrectly in their view, without explanation or alternative. Statistical analysis to show difference between proportions of counts of resistant vs all cultured colonies was done using Wilcoxon matched pairs signed-rank test (see lines 147-149, 240-243 and legends to table 4) and regression was only used for visualisation of a mean line (lines 135-137, 243-245, legends to figure 2). We have made this clearer. We have also consulted a second statistician who confirmed the suitability of these analyses both for visual and statistical purposes. They commented the following on the use of the Wilcoxon matched pairs signed-rank test: “The low power of non-parametric tests in general reinforces the strength of these highly significant results.”

Authors has not discussed which colonies they picked and no details were given on bacterial identification. 

>Added. Colonies were not picked, but counted. Colonies were not IDd but counted as Enterobacteriaceae based on shape, texture and colour (lines 130-135)

Data is not substantial to support the semi-quantitative quantification of normal flora.

> The reviewer is of the opinion that something was done incorrectly in their view, without explanation or alternative. 

We have added more information on methods, analysis and interpretation and believe that both the data and the consistent and significant differences between timepoints and drugs are sufficiently robust and substantial.

Figure S1 is about the seasonality of pathogens , I am confused, there is no discussion about seasonality. 

>Apologies, this sentence was cut out without making corrections to the figure. Text was restored and figure is referenced (lines 189-194)

It was discussed out of context in line # 151.

>See above. Apologies.

Among pathogens detected Strep. Pneumonia (98%), H.influenzae (12.3%) and rhinovirus (27%), somehow the discussion on rhinovirus was left out.

>We discussed why we did not include the bacteria in the analysis (lines 178-181, 322-341). We did not discuss all viral families separately, but have assessed appropriateness of antibiotic use taking viral PCR results into consideration for all viruses + atypical bacteria (lines 202-207)

Similarly, table 4 is hard to understand. The author should explain a little bit about the fraction of bacteria and antibiotics.

>We have tried to further clarify this, see lines 122-137, legends to figure 2 and table 4)

Add a sentence for a rationale of semi-quantitative detection and how well it represents normal intestinal flora? 

>Added, lines 137-140

It was not mentioned whether antibiotics were added in media or it was measured through disk diffusion?

>Antibiotics were added to the media. We described this more clearly (lines 122-137 and legends to figure 2 and table 4)

Molecular diagnostics: the details of methods should be given, with the name of the platform, principle of test and details of kits used.

>Added in lines 92-110

Overall, this manuscript requires language editing as sentence structure is not correct at various places and grammatical mistakes should be corrected.

>We thank the author for this comment and suggestion and have done our best to check and improve the manuscript for correct use of the English language 

various sentences donot make sense for example:

line 180-182 212-213, 221-222

>we have changed these and other sentences and hope they make sense to the reviewer now

Specific comments:

Abstract:

Objective: line# 3-5 needs to be rephrased

>Done (lines 3-7)

The word moleculary should be changed to molecular detection

>Changed “molecularly” to “using molecular assays” (line 10)

Line#38: The term GP stands for..?

>Clarified. GP stands for General Practitioner (line 40-41)

Line #131-139: data is not shown in any table,

>Correct, we do not feel repeating these numbers in a table would add significantly to the manuscript 

Line# 161: worse outcome should be defined if it is hospital discharge then analyses should be changed accordingly.

>Outcome was defined and recorded by the outpatient physicians (line 158-161). Unclear what the reviewer means with hospital discharge in the context of outpatient care. Hospital admission was a (rarely) recorded outcome (1.2%, line 161).

---

## [Decision Letter · Decision Letter 1]

5 Jun 2020

PONE-D-19-28321R1

Antibiotic use and prescription and its effects on intestinal flora in children with mild respiratory infections in Ho Chi Minh City, Vietnam. A prospective observational outpatient study.

PLOS ONE

Dear Dr. van Doorn,

Thank you for submitting your manuscript to PLOS ONE. In response to your request, your revised manscript was sent to 2 additional peer reviewers and I have now received feedback from both of them. After careful consideration, we feel that it has merit but does not fully meet PLOS ONE’s publication criteria as it currently stands. Therefore, we invite you to submit a revised version of the manuscript that addresses the points raised during the review process.

Please address the additional comments and in addition, it seems that Table 3b is missing from the manuscript.

We look forward to receiving your revised manuscript.

Kind regards,

Jane Foster, PhD

Academic Editor

PLOS ONE

Journal Requirements:

1.) Please ensure that your manuscript meets PLOS ONE's style requirements, including those for file naming. The PLOS ONE style templates can be found at http://www.plosone.org/attachments/PLOSOne_formatting_sample_main_body.pdf and http://www.plosone.org/attachments/PLOSOne_formatting_sample_title_authors_affiliations.pdf

2.) We noted in your submission details that a portion of your manuscript may have been presented or published elsewhere. "Validation of the HPLC detection methods of antibiotics in urine was described in 10.1002/bmc.4699 (als referenced in manuscript). The total percentage of positive samples (aggregated and without clinical metadata an d questionnaire results) was mentioned in this manuscript" Please clarify whether this [conference proceeding or publication] was peer-reviewed and formally published. If this work was previously peer-reviewed and published, in the cover letter please provide the reason that this work does not constitute dual publication and should be included in the current manuscript.

3.) In your Data Availability statement, you have not specified where the minimal data set underlying the results described in your manuscript can be found. PLOS defines a study's minimal data set as the underlying data used to reach the conclusions drawn in the manuscript and any additional data required to replicate the reported study findings in their entirety. All PLOS journals require that the minimal data set be made fully available. For more information about our data policy, please see http://journals.plos.org/plosone/s/data-availability.

4.) Please amend your list of authors on the manuscript to ensure that each author is linked to an affiliation. Authors’ affiliations should reflect the institution where the work was done (if authors moved subsequently, you can also list the new affiliation stating “current affiliation:….” as necessary).

We note that you have included affiliation numbers 1 - 10 however affiliations 8 does not have an author linked to it. Please amend affiliation 8 to link an author to it or remove if added in error.

5.) We note that you have included the phrase “data not shown” in your manuscript. Unfortunately, this does not meet our data sharing requirements. PLOS does not permit references to inaccessible data. We require that authors provide all relevant data within the paper, Supporting Information files, or in an acceptable, public repository. Please add a citation to support this phrase or upload the data that corresponds with these findings to a stable repository (such as Figshare or Dryad) and provide and URLs, DOIs, or accession numbers that may be used to access these data. Or, if the data are not a core part of the research being presented in your study, we ask that you remove the phrase that refers to these data.

6.) Please include your tables as part of your main manuscript and remove the individual files. Please note that supplementary tables should be uploaded as separate "Supporting Information" files.

7.) PRTC Notes I'm not sure if those comments also need to be added or it has already been sent back to author before.

8). Please provide additional details regarding participant consent. In the ethics statement in the Methods and online submission information, please ensure that you have specified (1) whether consent was informed and (2) what type you obtained (for instance, written or verbal). If your study included minors, state whether you obtained consent from parents or guardians. If the need for consent was waived by the ethics committee, please include this information.

Please also clarify whether your study was specifically reviewed and approved by your IRB.

2. Is the manuscript technically sound, and do the data support the conclusions?

Reviewer #2: Yes

Reviewer #3: Yes

3. Has the statistical analysis been performed appropriately and rigorously? 

Reviewer #2: Yes

Reviewer #3: Yes

4. Have the authors made all data underlying the findings in their manuscript fully available?

Reviewer #2: Yes

Reviewer #3: Yes

5. Is the manuscript presented in an intelligible fashion and written in standard English?

Reviewer #2: Yes

Reviewer #3: Yes

6. Review Comments to the Author

Reviewer #2: Antibiotic resistance is a global concern with one of the major contributors being the over-prescription of antibiotics. The current study shines a light on this issue by focusing on antibiotic use and prescription in children with mild respiratory infections in Ho Chi Minh City, Vietnam. To address this issue, the authors enrolled patients and recorded diagnoses, prescribed therapy and outcome at visit 1 and on follow-up after 7 days. Respiratory bacterial and viral pathogens were also detected using molecular assays. Antibiotic use before presentation was assessed using questionnaires and urine HPLC. The main finding reported is the overuse and over-prescription of antibiotics for uncomplicated respiratory infections.

This is an important topic and while the results are not surprising, they are still interesting and worth reporting. I have the following additional queries and recommendations:

(1) The extent to which these findings can be generalised to jurisdictions where antibiotics are available on prescription only needs to be considered more in the discussion. While the over-prescription aspect might be common, over the counter use varies quite a lot.

(2) I agree with the authors that an important part of this study is the measurement of antibiotics in urine. However, the sensitivity of the HPLC assay is very low at 49% and argues against the interpretation provided in the discussion that it may provide a more accurate and higher detection rate of antibiotic use than the questionnaire.

(3) Following on from this point, the authors also make a somewhat flawed leap in logic by stating that they can assume on the basis of this low sensitivity that antibiotic use is twice as high as detected in patients where no antibiotic use was reported. This needs to be removed from the discussion.

(4) The use of the term ‘gut flora’ is problematic as this description has been superseded by ‘gut microbiota’. In any case, since they have only assessed specific members in a targeted way, the analysis falls short of the current gold standards for a global assessment of the microbiota. This makes the title a little misleading and I suggest removing the reference to the gut flora altogether as it will create expectations that are not met with the analysis.

Reviewer #3: The authors of this paper have nicely showed the misuse of antibiotics in uncomplicated ARI in Vietnamese children. They have also determined the presence of different common respiratory viral and bacterial pathogens; as well as went on to determine the presence of antibiotics in their urine. Lastly they have also established the association of “selection of antibiotic-resistant members of Enterobacteriaceae in gut” with antibiotic use. I think that their data will further highlight the irrational overuse of antibiotics in ARI and its effect on the resistance development and selection in microbiota.

I am suggesting some minor edits to further improve the reader’s experience of this paper.

1. Though mentioned in abstract and results but please also mention the sample collection period in methods section.

2. Line 165-176 and lines 200-213. Under both these subheadings, though you have nicely defined results in the text form but would be nice if authors can also present their data in form of tables or figures, either in the paper or in supplementary information.

3. Lines 190-191. Please mention the whole virus name before using abbreviations. Similarly in the associated-figure also mention viral abbreviations.

4. Figure-2 was incorrectly marked as “Figure-1”.

7. PLOS authors have the option to publish the peer review history of their article (what does this mean?). If published, this will include your full peer review and any attached files.

Reviewer #2: No

Reviewer #3: No

---

## [Author Response · Author response to Decision Letter 1]

11 Oct 2020

Academic Editor

Point by point reply (marked with ">")

1.) Please ensure that your manuscript meets PLOS ONE's style requirements, including those for file naming. The PLOS ONE style templates can be found at http://www.plosone.org/attachments/PLOSOne_formatting_sample_main_body.pdf and http://www.plosone.org/attachments/PLOSOne_formatting_sample_title_authors_affiliations.pdf

>I have reformatted tables and figures and affiliations according to these requirements

2.) We noted in your submission details that a portion of your manuscript may have been presented or published elsewhere. "Validation of the HPLC detection methods of antibiotics in urine was described in 10.1002/bmc.4699 (als referenced in manuscript). The total percentage of positive samples (aggregated and without clinical metadata an d questionnaire results) was mentioned in this manuscript" Please clarify whether this [conference proceeding or publication] was peer-reviewed and formally published. If this work was previously peer-reviewed and published, in the cover letter please provide the reason that this work does not constitute dual publication and should be included in the current manuscript.

>We have added a paragraph on this in the cover letter:

Some data presented in this study have been published previously in a technical paper describing validation of the methods. This paper was peer reviewed and published in Biomedical Chromatography 34:e4699. This paper describes development of HPLC methods to detect antibiotics in urine and analysis of samples collected in our study. The paper deals almost entirely with technical and validation aspects of the methods and only has one paragraph and one figure (pasted in cover letter / response to reviewers files) where clinical evaluation and results on clinical samples are presented but only in an aggregated manner and without the context of clinical metadata and questionnaire results:

“The urine samples were collected from 563 pediatric patients under 16 years of age (50% of patients were less than 2 years old, and 95% of patients under 5 years old). The validated methods were successfully applied to determine the six β‐lactams in urine samples (10 patients with severe ARIs were anuria). Figure 2 presents the results of a qualitative measurement of the β‐lactams in clinical samples. Among of the tested β‐lactams, CFI was detected at the highest rate (54/553–9.8%), followed by amoxicillin (52/553–9.4%), while CFU was the least common identified medication, at only 1.8% (10/553).”

We reference the methods and their validation in the methods section of our paper and we present the results in context of their clinical metadata and the questionnaire results and therefore we believe this does not constitute double publication.

3.) In your Data Availability statement, you have not specified where the minimal data set underlying the results described in your manuscript can be found. PLOS defines a study's minimal data set as the underlying data used to reach the conclusions drawn in the manuscript and any additional data required to replicate the reported study findings in their entirety. All PLOS journals require that the minimal data set be made fully available. For more information about our data policy, please see http://journals.plos.org/plosone/s/data-availability.

>We have now uploaded our dataset with the submission

4.) Please amend your list of authors on the manuscript to ensure that each author is linked to an affiliation. Authors’ affiliations should reflect the institution where the work was done (if authors moved subsequently, you can also list the new affiliation stating “current affiliation:….” as necessary).

We note that you have included affiliation numbers 1 - 10 however affiliations 8 does not have an author linked to it. Please amend affiliation 8 to link an author to it or remove if added in error.

>Thank you for spotting this, this has been corrected

5.) We note that you have included the phrase “data not shown” in your manuscript. Unfortunately, this does not meet our data sharing requirements. PLOS does not permit references to inaccessible data. We require that authors provide all relevant data within the paper, Supporting Information files, or in an acceptable, public repository. Please add a citation to support this phrase or upload the data that corresponds with these findings to a stable repository (such as Figshare or Dryad) and provide and URLs, DOIs, or accession numbers that may be used to access these data. Or, if the data are not a core part of the research being presented in your study, we ask that you remove the phrase that refers to these data.

>The two occasions where we mentioned “data not shown” have been corrected. In the first instance about antibiotic regimen change we deleted the sentence, in the second instance on Streptococci and Haemophilus we deleted “(data not shown)” as the full data set has been uploaded now. 

6.) Please include your tables as part of your main manuscript and remove the individual files. Please note that supplementary tables should be uploaded as separate "Supporting Information" files.

>This has been corrected

7.) PRTC Notes I'm not sure if those comments also need to be added or it has already been sent back to author before.

>I have assumed there was nothing to address here

8). Please provide additional details regarding participant consent. In the ethics statement in the Methods and online submission information, please ensure that you have specified (1) whether consent was informed and (2) what type you obtained (for instance, written or verbal). If your study included minors, state whether you obtained consent from parents or guardians. If the need for consent was waived by the ethics committee, please include this information.

Please also clarify whether your study was specifically reviewed and approved by your IRB.

>This was all already included in the methods section, subsections “Patients and samples” and “Ethics approval”.

2. Is the manuscript technically sound, and do the data support the conclusions?

Reviewer #2: Yes

Reviewer #3: Yes

3. Has the statistical analysis been performed appropriately and rigorously?

Reviewer #2: Yes

Reviewer #3: Yes

4. Have the authors made all data underlying the findings in their manuscript fully available?

Reviewer #2: Yes

Reviewer #3: Yes

5. Is the manuscript presented in an intelligible fashion and written in standard English?

Reviewer #2: Yes

Reviewer #3: Yes

6. Review Comments to the Author

Reviewer #2: Antibiotic resistance is a global concern with one of the major contributors being the over-prescription of antibiotics. The current study shines a light on this issue by focusing on antibiotic use and prescription in children with mild respiratory infections in Ho Chi Minh City, Vietnam. To address this issue, the authors enrolled patients and recorded diagnoses, prescribed therapy and outcome at visit 1 and on follow-up after 7 days. Respiratory bacterial and viral pathogens were also detected using molecular assays. Antibiotic use before presentation was assessed using questionnaires and urine HPLC. The main finding reported is the overuse and over-prescription of antibiotics for uncomplicated respiratory infections.

This is an important topic and while the results are not surprising, they are still interesting and worth reporting. I have the following additional queries and recommendations:

(1) The extent to which these findings can be generalised to jurisdictions where antibiotics are available on prescription only needs to be considered more in the discussion. While the over-prescription aspect might be common, over the counter use varies quite a lot.

>We have added a sentence to the Limitations section on this

(2) I agree with the authors that an important part of this study is the measurement of antibiotics in urine. However, the sensitivity of the HPLC assay is very low at 49% and argues against the interpretation provided in the discussion that it may provide a more accurate and higher detection rate of antibiotic use than the questionnaire.

>We have rephrased this sentence

(3) Following on from this point, the authors also make a somewhat flawed leap in logic by stating that they can assume on the basis of this low sensitivity that antibiotic use is twice as high as detected in patients where no antibiotic use was reported. This needs to be removed from the discussion.

>We think that with the rephrasing of the previous sentence, this assumption / interpretation can still be made albeit with caution

(4) The use of the term ‘gut flora’ is problematic as this description has been superseded by ‘gut microbiota’. In any case, since they have only assessed specific members in a targeted way, the analysis falls short of the current gold standards for a global assessment of the microbiota. This makes the title a little misleading and I suggest removing the reference to the gut flora altogether as it will create expectations that are not met with the analysis.

>We have removed this from the manuscript and instead refer to this work using “intestinal Enterobacteriaceae”

Reviewer #3: The authors of this paper have nicely showed the misuse of antibiotics in uncomplicated ARI in Vietnamese children. They have also determined the presence of different common respiratory viral and bacterial pathogens; as well as went on to determine the presence of antibiotics in their urine. Lastly they have also established the association of “selection of antibiotic-resistant members of Enterobacteriaceae in gut” with antibiotic use. I think that their data will further highlight the irrational overuse of antibiotics in ARI and its effect on the resistance development and selection in microbiota.

I am suggesting some minor edits to further improve the reader’s experience of this paper.

1. Though mentioned in abstract and results but please also mention the sample collection period in methods section.

>Sorry to disagree here, but I don’t think the dates of the actual delivery of methods should be in the methods, as it is a result in my opinion. Happy to add if the editor / reviewer insists, but I don’t think it belongs in methods.

2. Line 165-176 and lines 200-213. Under both these subheadings, though you have nicely defined results in the text form but would be nice if authors can also present their data in form of tables or figures, either in the paper or in supplementary information.

>We have now included all figures/numbers from these sections in the tables

3. Lines 190-191. Please mention the whole virus name before using abbreviations. Similarly in the associated-figure also mention viral abbreviations.

>This has been corrected

4. Figure-2 was incorrectly marked as “Figure-1”.

>Apologies, this has been corrected

---

## [Editor Report · Decision Letter 2]

21 Oct 2020

Antibiotic use and prescription and its effects on Enterobacteriaceae in the gut in children with mild respiratory infections in Ho Chi Minh City, Vietnam. A prospective observational outpatient study.

PONE-D-19-28321R2

Dear Dr. van Doorn,

We’re pleased to inform you that your manuscript has been judged scientifically suitable for publication and will be formally accepted for publication once it meets all outstanding technical requirements.

Kind regards,

Jane Foster, PhD

Academic Editor

PLOS ONE

---

## [Editor Report · Acceptance letter]

23 Oct 2020

PONE-D-19-28321R2 

Antibiotic use and prescription and its effects on *Enterobacteriaceae* in the gut in children with mild respiratory infections in Ho Chi Minh City, Vietnam. A prospective observational outpatient study. 

Dear Dr. van Doorn:

I'm pleased to inform you that your manuscript has been deemed suitable for publication in PLOS ONE. Congratulations! Your manuscript is now with our production department. 

Kind regards, 

on behalf of

Dr. Jane Foster 

Academic Editor

PLOS ONE